# Small Molecule Cocktails Promote Fibroblast-to-Leydig-like Cell Conversion for Hypogonadism Therapy

**DOI:** 10.3390/pharmaceutics15102456

**Published:** 2023-10-13

**Authors:** Fei Yuan, Kaiping Bai, Yanping Hou, Xiangyu Zou, Jie Sun

**Affiliations:** Department of Urology, Shanghai Children’s Medical Center, Shanghai Jiao Tong University of Medicine, No. 1678 Dongfang Road, Pudong New Area, Shanghai 200127, China; yf13072381838@sjtu.edu.cn (F.Y.); bkp09024@sjtu.edu.cn (K.B.); houyanping@scmc.com.cn (Y.H.); zouxiangyu@scmc.com.cn (X.Z.)

**Keywords:** small molecule cocktails, fibroblast, Leydig-like cell, male hypogonadism

## Abstract

Male hypogonadism arises from the inadequate production of testosterone (T) by the testes, primarily due to Leydig cell (LC) dysfunction. Small molecules possess several advantages, including high cell permeability, ease of synthesis, standardization, and low effective concentration. Recent investigations have illuminated the potential of small molecule combinations to facilitate direct lineage reprogramming, removing the need for transgenes by modulating cellular signaling pathways and epigenetic modifications. In this study, we have identified a specific cocktail of small molecules, comprising forskolin, DAPT, purmorphamine, 8-Br-cAMP, 20α-hydroxycholesterol, and SAG, capable of promoting the conversion of fibroblasts into Leydig-like cells (LLCs). These LLCs expressed key genes involved in testosterone synthesis, such as Star, Cyp11a1, and Hsd3b1, and exhibited the ability to secrete testosterone in vitro. Furthermore, they successfully restored serum testosterone levels in testosterone-castrated mice in vivo. The small molecule cocktails also induced alterations in the epigenetic marks, specifically H3K4me3, and enhanced chromosomal accessibility on core steroidogenesis genes. This study presents a reliable methodology for generating Leydig-like seed cells that holds promise as a novel therapeutic approach for hypogonadism.

## 1. Introduction

Male hypogonadism arises when the testes fail to produce sufficient testosterone (T) due to Leydig cell (LC) dysfunction [1]. A comprehensive study involving over 6000 Chinese men reported that 8% of middle-aged and elderly individuals exhibited hypogonadism, as indicated by low serum T levels and associated sexual symptoms [2]. While pediatric hypogonadism is relatively rare, it can give rise to a spectrum of issues, including abnormal genitalia, delayed puberty, undescended testes, and metabolic syndrome, with implications for sexual and psychological development during youth and quality of life in adulthood [3,4].

The primary therapeutic approach for this condition is testosterone replacement therapy (TRT), which aims to restore normal testosterone levels and enhance physical function. However, TRT carries potential side effects and safety concerns, including thromboembolism, male pattern baldness, and an increased risk of prostate cancer [5,6,7]. Additionally, exogenous hormone therapy fails to replicate the natural feedback mechanisms of the hypothalamic-pituitary-gonadal axis, leading to inconsistencies with patients’ circadian rhythms and physiological fluctuations [8,9]. An attractive alternative for male hypogonadism is the transplantation of LCs or testosterone-producing Leydig-like cells (LLCs), offering a solution without the drawbacks associated with TRT [9,10]. However, the limited availability of LCs restricts its clinical application. Previous studies in our laboratory have successfully generated LLCs capable of testosterone production from human foreskin fibroblasts (HFFs) using genetic methods [11,12]. Nevertheless, the potential risk of tumorigenesis associated with transgenic approaches warrants careful consideration [13].

Small molecules, with a molecular weight below 900 Da, possess nanoscale dimensions and exhibit advantageous properties, including non-immunogenicity, high cell permeability, ease of synthesis, and standardization. Recent investigations have unveiled the potential of small molecule combinations to drive direct lineage reprogramming without relying on transgenes by modulating cellular signaling pathways and inducing epigenetic modifications [14,15,16]. Notably, our recent achievements include the successful conversion of HFFs into LLCs using small molecular compounds in conjunction with the overexpression of the nuclear receptor subfamily 5 group A member 1 (NR5A1), a pivotal transcription factor involved in steroid production [17,18]. In this study, our objective was to pinpoint specific small molecular compounds capable of transforming fibroblasts into LLCs, as substantiated by transcriptome analysis. Additionally, we investigated alterations in fibroblast epigenetic marks, such as H3K4me3, following treatment with these small molecules. This research presents a safe and dependable avenue for obtaining LLCs for transplantation, offering a novel therapeutic approach for male hypogonadism.

## 2. Materials and Methods

### 2.1. Isolation and Culture of HFFs

HFFs were obtained from foreskin specimens of healthy pediatric donors, collected by Shanghai Children’s Medical Center. The isolation and cultivation of HFFs were conducted following established procedures [12]. Briefly, the specimens underwent triple washing with phosphate-buffered saline (PBS, Gibco™, Waltham, MA, USA, Cat. No: 10010023) supplemented with 1% penicillin-streptomycin (P/S, Gibco™, Cat. No: 15140122). Subsequently, subcutaneous connective tissues were excised, and the remaining samples were cut into pieces measuring 0.5 to 1.0 cm. These fragments were then incubated in a solution containing 1 mg/mL collagenase IV (Gibco™, Cat. No: 17104019) for 2 h at 37 °C. After digestion, the solution was filtered through a 100 um nylon mesh and centrifuged at 1500 rpm for 5 min. The supernatant was discarded, and the cell pellet was resuspended and cultured in DMEM/F12 medium (HyClone, Logan, UT, USA, Cat. No: 11320033) supplemented with 10% fetal bovine serum (FBS, Gibco™, Cat. No: 10099141C) under conditions of 5% CO_2_ at 37 °C. HFFs were utilized at passages 3 to 5.

### 2.2. Isolation and Culture of MEFs

Mouse embryo fibroblasts (MEFs) were derived from Balb/c mouse embryos at E12.5–13.5. The embryos’ head and visceral tissues were carefully removed, and the remaining tissues were minced with forceps. Subsequently, the tissue fragments were incubated in a solution containing 0.25% trypsin and 1 mM EDTA (NCM Biotech, Suzhou, China, Cat. No: C125C1) for 10–15 min at 37 °C. After terminating the digestion, the suspension was left undisturbed for 15 min, and the upper cell suspension was collected. This suspension was then filtered through a 100-um nylon mesh and centrifuged at 1500 rpm for 5 min. The cell pellet was resuspended and cultured in DMEM medium (Gibco™, Waltham, MA, USA, Cat. No: 11995065) supplemented with 10% FBS, 2 mM L-glutamine, and 1% P/S. Cells were seeded in 90 mm dishes and cultivated in an incubator at 5% CO_2_ and 37 °C. After 4–6 h, nonadherent cells were discarded, and adherent MEFs were cultured until confluence was achieved. The medium was replaced every 2 days, and MEFs were used at passages 3 to 5.

### 2.3. Isolation and Culture of PLCs

Primary Leydig cells (PLCs) were obtained from adult male mice purchased from Shanghai Jihui Experimental Animal Breeding Co., Ltd. (Shanghai, China, 8 weeks old). Decapsulated testes were enzymatically dispersed with trypsin in DMEM for 15 min at 37 °C. The dispersed cells were filtered through a 40 um nylon mesh, and cells with densities of 1.070 g/mL and higher were collected following centrifugation at 23,500× *g* for 45 min at 4 °C. Isolated PLCs were seeded in six-well plates and cultured in DMEM containing 10% FBS at 37 °C.

### 2.4. Generation of LLCs

For the generation of LLCs, the time point at which MEFs and HFFs (1 × 10^5^ cells/well) were plated into 6-well dishes and cultured with standard medium was designated as day 0 (D0). During the first differentiation stage, spanning from day 1 (D1) to day 4 (D4), the medium was substituted with a 4C inducing medium containing 10 μM CHIR99021, 2 μM SB431542, 10 μM EPZ004777, and 0.2 μM SAG. In the second differentiation stage, extending from day 5 (D5) to day 19 (D19), a 6C induction medium comprising 10 μM forskolin, 1 μM DAPT, 1 μM purmorphamine, 0.2 μM SAG, 20α-hydroxycholesterol (0.2 μM), and 8-Br-cAMP (1 μM) replaced the previous medium. Subsequently, LLCs were cultured in a Leydig cell medium with DMEM containing 5% horse serum. The culture medium was refreshed every 2 days. The specific working concentrations and information for all small molecule compounds mentioned in this study are provided in Table 1.

### 2.5. Immunofluorescence Assays

To conduct immunofluorescence assays, cell specimens were initially fixed using 4% paraformaldehyde (Sangon Biotech, Shanghai, China, Cat. No: A500684) for 15 min. Subsequently, cells were subjected to three washes with PBS. The cells were then permeabilized with 0.1% Triton X-100 (Sangon Biotech, Cat. No: A600198) in PBS for 15 min at room temperature and incubated in 3% BSA (Beyotime Biotech, Nantong, China, Cat. No: ST023) in PBS for 1 h at room temperature. Next, the cells were incubated overnight at 4 °C with primary antibodies, followed by incubation with fluorescein isothiocyanate-conjugated anti-mouse or anti-rabbit IgG secondary antibodies for 60 min at room temperature. Detailed product information for the corresponding primary and secondary antibodies is provided in Appendix A. Following antibody incubation, the cells were washed three times with PBS for 5 min each and subsequently incubated for 5 min with DAPI for nuclear staining. After a final three washes with PBS, the specimens were examined using an inverted fluorescence microscope.

### 2.6. RNA-Seq

RNA sequencing was performed on collected cells. Three samples from each group were subjected to RNA isolation using the RNeasy Micro Kit (Qiagen, Hilden, Germany, Cat. No: 74004) according to the manufacturer’s instructions. Subsequently, libraries were constructed, and RNA sequencing was carried out by Zhejiang Annoroad Biotechnology Co., Ltd. (Beijing, China). Sequenced reads were mapped to the mouse genome GRCm39 using HISAT2 (version 2.2.1.0). Raw counts were generated and quantified using HTSeq-Count (version 0.9.1). Differential gene expression analysis was conducted using DESeq2 (version 1.20.0) and DESeq (version 1.18.0). Differentially expressed genes (DEGs) were selected based on a combination of a fold change threshold ≥ 2 and an FDR (*q*-Value) threshold ≤ 0.05.

### 2.7. RNA Extraction and Quantitative Real-Time PCR (qRT-PCR)

Total RNAs were extracted using Trizol reagent (Invitrogen™, Waltham, MA, USA, Cat. No: 15596018CN) and subsequently reverse-transcribed into cDNA using the PrimeScript™ RT reagent Kit (Takara Biotech, Kusatsu, Japan, Cat. No: RR037A). Quantitative real-time PCR (qRT-PCR) was performed using specific primers and PowerUp™ SYBR™Green Master Mix (Applied Biosystems™, Waltham, MA, USA, Cat. No: A25742) on a C1000 Touch Thermal Cycler with a CFX96 real-time system. Expression data were normalized to the Gapdh or β-actin level, and each qRT-PCR reaction was conducted in triplicate. Primer information is provided in Appendix A.

### 2.8. Western Blot

Protein samples were applied to a 10% sodium dodecyl sulfate polyacrylamide gel electrophoresis and transferred onto polyvinylidene difluoride membranes using an electroblot apparatus. After blocking with a solution containing 5% fat-free milk (Beyotime Biotech, Cat. No: P0216) for 1 h at room temperature, the membranes were incubated overnight at 4 °C with primary antibodies. Following three washes with tris-buffered saline containing Tween 20 (TBST), the membranes were incubated for 1 h at room temperature with a horseradish peroxidase-conjugated secondary antibody (1:5000). Subsequently, the membranes underwent three additional TBST washes. Visualization of protein bands was achieved using ECL-enhanced chemiluminescence (Yeasen Biotech, Shanghai, China, Cat. No: 36208ES60). Primary and secondary antibody information is provided in Appendix A.

### 2.9. Testosterone Measurement

Testosterone concentration was determined using the Access Testosterone Kit (R&D Systems, Santa Clara, CA, USA, Cat. No: KGE010), as previously employed in our study. Briefly, serum or culture supernatants collected at specified time points were used for measurement. The UniCel DxI 800 system (Beckman Coulter, Brea, CA, USA) automatically conducted the measurements, with a detection limit ranging from 0.1 to 10 ng/mL, and both intra- and inter-assay variations were below 5% and 10%, respectively.

### 2.10. LM-CS Analyses

On day 21 of the induction process, culture media from both LLCs and mice Leydig cells were collected. Steroid hormones, including progesterone, DHEA, testosterone, cortisol, and estrogen, were simultaneously assayed using ultra-high-performance liquid chromatography coupled with triple quadrupole mass spectrometry (UHPLC-MS/MS). Specifically, 120 μL of each sample’s media was mixed with 240 μL of methanol and 90 μL of a steroid solution containing ethinylestradiol (450 pg/mL) and norethisterone (1 ng/mL). Data were collected in centroid mode and processed using Masslynx 4.1 software and the Quanlynx program (Version 4.1).

### 2.11. Animals()

At six weeks of age, Balb/c mice were obtained from Shanghai Jihui Experimental Animal Breeding Co., Ltd. The animals were maintained in controlled conditions with a temperature of 23 ± 2 °C, a 12-h dark/light cycle, and 45–55% relative humidity. All surgical procedures and postoperative care were conducted in compliance with the guidelines outlined in the Guide for the Care and Use of Laboratory Animals, as approved by the Animal Care and Use Committee of Shanghai Children’s Medical Center.

### 2.12. Cell Tracking

For cell tracking, 2–4 × 10^6^ cells were incubated with PKH 26 (MCE, Monmouth Junction, NJ, USA, Cat. No: HY-D1451) for 24 h and subsequently digested with trypsin. Cells were washed twice with sterile PBS to remove excess dye and then centrifuged before being placed on ice for cell transplantation.

### 2.13. Castration and Cell Transplantation

At six weeks of age, Balb/c mice were randomly divided into four groups: the control group (5 mice), model group (5 mice), HFFs group (5 mice), and LLCs group (5 mice). To investigate the effect of LLCs on serum testosterone levels in castrated mice, we induced castration models following previously established procedures [9]. Briefly, isoflurane was administered at a rate of 300–500 mL/min to anesthetize the mice. After ensuring proper hemostasis, we removed parenchymal tissue from both testes through an albuginea incision. Subsequently, we washed out any remaining blood in the fibrous capsule using PBS and closed the albuginea incision with sutures. Next, we injected 0.2 mL of cold liquid Matrigel (Corning, Corning, NY, USA, Cat. No: 354230) containing approximately 1 × 10^6^ transplanted LLCs and HFFs labeled with PKH 26 into the fibrous capsule of the recipient testis, while the model group received an equal volume of liquid Matrigel without cells. Testosterone or Matrigel implants in all groups were assessed one week after cell transplantation.

### 2.14. Cut-Tag Analyses

Cleavage Under Targets & Tagmentation sequencing (CUT&Tag-seq) libraries were prepared using the Hyperactive Universal CUT&Tag Assay Kit for Illumina (Vazyme, Nanjing, China, Cat. No: TD903-01) following the manufacturer’s instructions. Briefly, 2 × 10^5^ cells were bound to ConA beads and incubated at room temperature for 2 h with a primary antibody against H3K4me3 (diluted 1:50 in primary antibody buffer). The beads were washed several times in wash buffer, followed by incubation at room temperature for 1 h with a goat anti-rabbit IgG secondary antibody (diluted 1:100 in antibody buffer). After additional washings in wash buffer, the beads were incubated at room temperature for 1 h with pAG-Tn5 transposon (diluted 1:50 in Dig-300 buffer). The mixture was then subjected to incubation at 37 °C for 1 h, followed by termination of tagmentation through heating at 55 °C for 10 min. DNA fragments were purified using DNA extract beads and subjected to PCR amplification. The PCR reactions contained 15 μL of extracted DNA, 5 μL of primer i5 (N501), 5 μL of primer i7 (N701-N706), and 25 μL of 2× CAM buffer (Vazyme, Cat. No: TD202). The amplification consisted of 9~20 cycles, depending on cell numbers. PCR products were subsequently purified using VAHTS DNA Clean Beads (Vazyme, Cat. No: N411-01). Library sequencing and data analysis were performed, involving the sequencing of libraries on an Illumina^®^ platform, filtering of raw reads, mapping to the mouse genome GRCm39 using bowtie2 (version 2.2.9), peak calling using SEACR (version 1.3), and annotation using ChIPseeker (version 1.12.1). Details regarding the primary and secondary antibodies used can be found in Appendix A.

### 2.15. Statistical Analyses

The data are presented as the mean ± one standard deviation around the mean (SD), with each experiment conducted at least three times. For comparisons among more than two groups, statistical analyses were performed using either a one-way ANOVA or an unpaired Student’s *t*-test. Statistical significance was considered when the two-tailed *p*-value was less than 0.05.

## 3. Results

### 3.1. Transcriptome Analysis Reveals Distinct Signaling Pathways between Fibroblasts and LCs

We isolated mouse embryonic fibroblasts (MEFs) and primary Leydig cells (PLCs) from adult mice. To confirm the purity of these cell populations, we conducted immunofluorescence analysis of vimentin (a marker protein for fibroblasts) and cyp11a1 (a marker for LCs). The expression rates of these markers reached 99% and 96%, respectively (Figure 1A). Subsequently, we conducted transcriptome profiling of MEFs and PLCs using RNA-Seq. We identified 4097 DEGs, including 2724 upregulated and 1373 downregulated genes. Heatmaps and volcano plots of DEGs are presented in Figure 1B and Figure 1C, respectively. DEGs were significantly enriched in Gene Ontology (GO) terms related to signal transduction and signaling receptor binding (Appendix A). KEGG enrichment analysis of DEGs revealed significant enrichment in pathways such as PI3K-Akt, calcium, cAMP, Rap1, cGMP−PKG, steroid synthesis, and metabolism of xenobiotics by cytochrome P450, aldosterone synthesis and secretion, and cortisol synthesis and secretion (Figure 1D). Additionally, a protein–protein network interaction (PPI) network, integrated with the KEGG regulation map, revealed the important connections between Leydig function-related genes and signaling pathways (Figure 2). These results highlight the presence of distinct signaling pathways between fibroblasts and LCs, underscoring their close relationship with the steroid synthesis function.

### 3.2. Screen out the Combination of Small Molecule Compounds That Activate NR5A1 Expression in Fibroblasts

We conducted a screening for small molecules targeting the regulatory signaling pathways enriched in the RNA-seq data. Simultaneously, we considered small molecules that have previously played important roles in LC lineage differentiation as candidates for converting fibroblasts into LLCs in this study. Ultimately, we selected 11 small molecule compounds (11C) targeting various signaling pathways and epigenetic modifications (listed in Table 1). Subsequently, we treated HFFs and MEFs with these molecules for 14 days. However, neither cell type showed significant changes in LC lineage markers, and no testosterone was detected in the cell culture medium. Consequently, we adjusted our induction strategy, as illustrated in Figure 3A. Initially, we added a combination of four compounds (4C) composed of CHIR99021, SB431542, EPZ004777, and SAG to the culture as these four small molecules were previously reported to facilitate early chemical reprogramming of fibroblasts. This treatment downregulated fibroblast marker genes Twist1 and Snail2 while upregulating epithelial cell-related genes Krt8 and Krt18 (Figure 3B), suggesting that it promoted mesenchymal-epithelial transition (MET), a process known to enhance fibroblast inducibility [19]. However, there was no significant difference in Nr5a1 gene expression (Appendix A). Subsequently, we introduced the 11C into the medium and analyzed its impact on Nr5a1 expression. We found that the 11C effectively activated Nr5a1 expression after 7 days of treatment. To identify the most effective combination, we screened out individual molecules from the 11C. The results indicated that removal of forskolin, DAPT, purmorphamine, 8-Br-cAMP, 20α-hydroxycholesterol, or SAG resulted in decreased Nr5a1 gene expression (Figure 3C). Notably, the combined action of these six compounds (6C) significantly enhanced Nr5a1 expression at both mRNA and protein levels (Figure 3D–E).

### 3.3. Convert Fibroblast into LLCs by Small Molecular Compounds In Vitro

While the 6C combination effectively initiated Nr5a1 expression at 7 days, several LC marker genes, especially those related to testosterone synthesis, exhibited increased expression that was not statistically significant. However, when we extended the induction period to 14 days, mRNA expression levels of LC marker genes, including Star, Cyp11a1, Hsd3b1, and Hsd17b3, notably increased (Figure 3F). The Western blot confirmed positive expression of STAR, CYP11A1, and 3β-HSD (Figure 3G). Immunofluorescence assays also confirmed widespread expression of HSD3B1 in the 6C-treated group, whereas these proteins were not detected in the control group (Figure 3H).

Moreover, we detected testosterone levels in the cell medium supernatant using the Testosterone Assay Kit. Cells in the 6C treatment group secreted a modest amount of testosterone. After 30 min of stimulation with 10 ng/mL LH, testosterone levels in the LLCs group were significantly higher than in the control group (Figure 4A). To further identify the types of steroid hormones secreted by cells in the 6C group, we performed LC-MS to compare the types of steroid hormones produced between MEFs treated with 6C and mice PLCs. We found that cells in the 6C group produced not only testosterone but also some cortisol. Estrogen production was not observed in the LLCs. Additionally, the two main types of hormones produced by this group were pregnenolone and progesterone, both of which are precursor steroids for testosterone synthesis. In contrast, the hormones secreted by the PLCs were dominated by DHEA and testosterone and did not include corticosteroids or estrogen (Figure 4B). Although cells treated with 6C secreted different hormone products than PLCs, they exhibited testosterone production capacity and responsiveness to LH (Figure 4A). Furthermore, following 14 days of treatment with the 6C induction, the morphology of chemically induced HFFs changed. The cell edges gradually transitioned from radial and irregular to smooth or oval, with concentrated nucleoli (Figure 4C).

Similar results were observed in the MEFs, which were induced by the small-molecule cocktails for 21 days. The morphology of the MEFs also transitioned to a smooth or oval shape (Appendix A). The expression of the Nr5a1 gene and key genes related to testosterone synthesis, including Cyp11a1, Hsd3b1, Hsd17b3, were significantly increased (Appendix A). Testosterone levels were also detected in the culture medium (Appendix A). These findings suggest that we successfully converted fibroblasts into LLCs using defined small molecule cocktails in vitro.

### 3.4. Verify the Therapeutic Effect of LLCs Transplantation on Testosterone Castration In Vivo

To investigate whether LLCs could survive and produce testosterone in vivo as functional LC, we created a mouse testosterone castration model by removing testicular contents. The LLCs were labeled with PKH 26 for tracking, and we injected 200 μL of extracellular Matrigel containing PKH 26-labeled cells (1 × 10^6^/mL) into the tunica albuginea of castrated mice. HFFs without 6C treatment were used as a negative control (Figure 4D). Serum testosterone levels were analyzed after 7 days of transplantation. The results showed that serum testosterone levels significantly declined in the castrated group and the HFFs group. However, in castrated mice with LLC implantation, serum testosterone levels were lower than those in the normal group but significantly higher than in the two groups mentioned above (Figure 4E). Moreover, mouse testes were collected for immunofluorescence staining, which demonstrated that the LLCs still expressed the testosterone-synthesizing enzyme CYP11A1 after 7 days of transplantation (Figure 3F). These results indicate that the LLCs expressed testosterone-synthesizing enzymes and that LLC transplantation effectively promoted the recovery of serum testosterone levels in testis-castrated mice in vivo.

### 3.5. Fibroblast Altered the Epigenetic Modification of H3K4me3 by Small Molecule Cocktails

As small molecule compounds induce somatic reprogramming by altering intracellular signaling and cellular epigenetic modifications, we hypothesized that our small molecule cocktails transformed fibroblasts into LLCs by promoting an epigenetic state characterized by open chromatin. We compared the levels of H3K4me3 before and after small molecule induction and observed a significant increase in H3K4me3 using Western blot (Figure 5A). To further investigate this mechanism, we analyzed genome-wide epigenetic changes at D0 and D21 using cut-tag analysis of H3K4me3, an active chromatin marker. The analysis indicated that H3K4me3 patterns significantly decreased at gene promoters and transcription start sites (TSS) (Figure 5B). Peaks of H3K4me3 from the second replicate were primarily enriched in promoter regions (Figure 5C). KEGG enrichment analysis of the differential genes revealed significant enrichment of upregulated genes in drug metabolism, cytochrome P450, Cushing syndrome, cortisol synthesis and secretion, and Rap1 signaling pathways (Figure 5D). Collectively, our analyses suggest that small molecule cocktails induce epigenetic remodeling, promoting the conversion of the passive chromatin state of fibroblasts into a more euchromatic state. This enhances chromatin accessibility at core steroid synthesis gene loci, facilitating the transition to Leydig cells.

## 4. Discussion

In this study, we have demonstrated the feasibility of converting fibroblasts, especially HFFs, into LLCs using small molecule cocktails. These LLCs not only expressed LC marker genes and produced testosterone in vitro but also restored serum testosterone levels in hypogonadal mice in vivo. The small molecule cocktails induced significant epigenetic changes in H3K4me3 in fibroblasts, increasing chromatin accessibility at core steroid synthesis gene loci and facilitating their transition into Leydig cells.

Previous research has reported the conversion of various cell types, such as mesenchymal stem cells (MSCs), embryonic stem cells (ESCs), and induced pluripotent stem cells (iPSCs), into steroid-generating cells using molecular compounds or transcription factors [20,21,22,23,24]. However, iPSCs/ESCs-derived cells and transgenic cells have ethical and safety concerns for clinical applications [12]. Moreover, generating LLCs from small molecules in previous studies was inefficient, complicated, and time-consuming. HFFs, originating from patients with hypogonadism, are easily obtainable and do not pose immunogenicity issues. Our prior experiments have shown that HFFs can be converted into LLCs using a transcription factor combined with small molecular compounds [17]. Additionally, there is research indicating that small molecules can transform MEFs into LLCs [24]. However, these methods did not work for HFFs. Therefore, we optimized the small molecule cocktails to convert fibroblasts, especially HFFs, into LLCs in the current study.

By comparing the gene expression profiles of MEFs and PLCs, we identified critical associations between LC marker gene expression and signaling pathways, including DHH, cAMP, cGMP−PKG, and Rap1. DHH, secreted by Sertoli cells (SCs) in the testes, is essential for LC development. Rat testes with a null DHH gene mutation lacked adult LCs [25]. Activation of the cAMP/PKA cascade significantly increased Star expression and genes related to steroidogenesis in mouse Leydig cells [26]. The cGMP−PKG signaling pathway can contribute to the recovery of Leydig cell testosterone production under stress stimulation [27]. Thus, we screened for small molecular compounds targeting these pathways, which were enriched in the transcriptome profiles of MEFs and PLCs. We also considered small molecules that had previously been reported to induce other cell types into LLCs.

In our study, during the first stage of induction from day 1 to 4, we introduced CHIR99021, SB431542, EPZ004777, and SAG into the culture medium. CHIR99021 inhibits GSK-3β, leading to Wnt pathway activation. Activation of the Wnt signaling pathway has been reported to improve the efficiency of induction and sustain stem cell self-renewal [28]. EPZ004777, as an epigenetic regulator, displays remarkable selectivity for inhibiting the histone H3K79 methyltransferase, DOT1L. Previous research indicates that EPZ004777 significantly improved reprogramming efficiency during the generation of mouse iPSCs [29]. SB431542 is an inhibitor of the TGF-β signaling pathway. The TGF-β signaling pathway inhibits MET while promoting EMT, which is detrimental to reprogramming [30]. Accumulating evidence suggests that MET is essential during early somatic cell reprogramming. Inhibition of TGF-β signals by SB-431542 promotes fibroblast reprogramming. Furthermore, the TGF-β pathway is associated not only with cell fibrosis but also negatively regulates steroidogenic gene expression and LC lineage differentiation in the testis [31,32]. SAG promotes the secretion of laminin, assisting in LC proliferation [33]. All four of these small molecule compounds were used in the early stages of fibroblast reprogramming. Taken together, CHIR99021, SB431542, EPZ004777, and SAG facilitated fibroblast MET in our study. During the first stage of induction, these four compounds induced chromosomal changes in fibroblasts, making it easier for them to transition into Leydig cells.

In the second stage of differentiation from day 5 to 19, we identified small molecules that could induce fibroblasts to express NR5A1. NR5A1 is essential for maintaining the steroidogenic function of Leydig cells. Knockout of NR5A1 results in Leydig cells losing the ability to generate steroid hormones despite their presence in the embryonic testis [34]. We screened small molecules targeting cell signaling pathways that promote Leydig cell differentiation. As a result, we identified 6C, which includes forskolin, DAPT, purmorphamine, SAG, 20α-hydroxycholesterol, and 8-Br-cAMP, as small molecules that significantly activated NR5A1 expression within 7 days. Forskolin activates the cAMP/PKA signaling pathway, and 8-Br-cAMP is a cAMP analog. The cAMP/PKA pathway mediates cholesterol transport into mitochondria and activates the protein expression of enzymes related to steroidogenesis. DAPT is a Notch pathway inhibitor. The Notch pathway prevents Leydig stem cell differentiation into mature LCs [34,35]. Purmorphamine and SAG are both Hedgehog pathway activators. Purmorphamine’s effects are induced by the stimulation of the signaling cascade and an increase in the expression of genes such a Gli1, Gli2, Ptch1, and Ptch2 [35]. SAG has been reported to induce stem cells like MSCs into Leydig-like cells. 20α-hydroxycholesterol, a membrane-permeable cholesterol analog, has been reported as a novel activator of the Hedgehog pathway that influences the osteogenic and adipogenic differentiation of MSCs [36,37].

LLCs generated through the method described above can express enzymes related to testosterone synthesis and produce testosterone. However, the steroid hormone products of LLCs differ from those of LCs. LLCs primarily produce precursor hormones of testosterone, such as progesterone and pregnenolone. Under physiological conditions, LCs synthesize testosterone from cholesterol through a series of steps, including LH binding to its receptor LHCGR, cAMP production, cholesterol translocation from the cytoplasm into mitochondria, conversion of cholesterol into pregnenolone by CYP11A1, and subsequent transformation of pregnenolone into testosterone [38]. These findings demonstrate that the LLCs we generated can efficiently express CYP11A1 but exhibit lower efficiency in activating HSD17B3. We intend to optimize this method in future studies to enhance HSD17B3 expression. Additionally, we observed a significant decrease in the cell proliferation rate with extended induction time (Figure 4C and Appendix A). Two possible causes for this phenomenon are apparent: one is that actively proliferating fibroblasts have transitioned into less proliferative Leydig cells due to the induction by cocktails, and the other is the potential cytotoxic effects of one or several of the cocktails. To determine the underlying reason, we examined the impact of each drug in the cocktails on cell proliferation. The results indicate that the concentrations of most molecules used had minimal cytotoxic effects on fibroblasts within 7 days, with the exception of purmorphamine and 20α-hydroxycholesterol (Appendix A).

To investigate the mechanism by which small molecule cocktails induce fibroblasts to become Leydig cells, we explored the changes in epigenetic modifications of H3K4me3 in fibroblasts induced by these cocktails. H3K4me3, as a hallmark of transcript activation, reflects the open chromatin state at the target gene loci. Small molecule cocktails significantly altered H3K4me3 enrichment in the promoter regions of core steroid synthesis genes, thereby enhancing the efficiency of conversion into LCs. This research represents the first report of alterations in H3K4me3 modifications during the process of converting fibroblasts into Leydig cells induced by small molecular cocktails. We identified that small molecule cocktails upregulated H3K4me3 enrichment at target genes, especially those related to steroid synthesis, promoting the expression of relevant genes.

In conclusion, we have developed a novel differentiation protocol to induce fibroblasts, especially HFFs, into testosterone-producing LLCs using defined molecular cocktails, without introducing any exogenous transcription factor genes. Our findings may offer new insights into Leydig cell transplantation therapy for testosterone deficiency or decline.

## Figures and Tables

**Figure 1 pharmaceutics-15-02456-f001:**
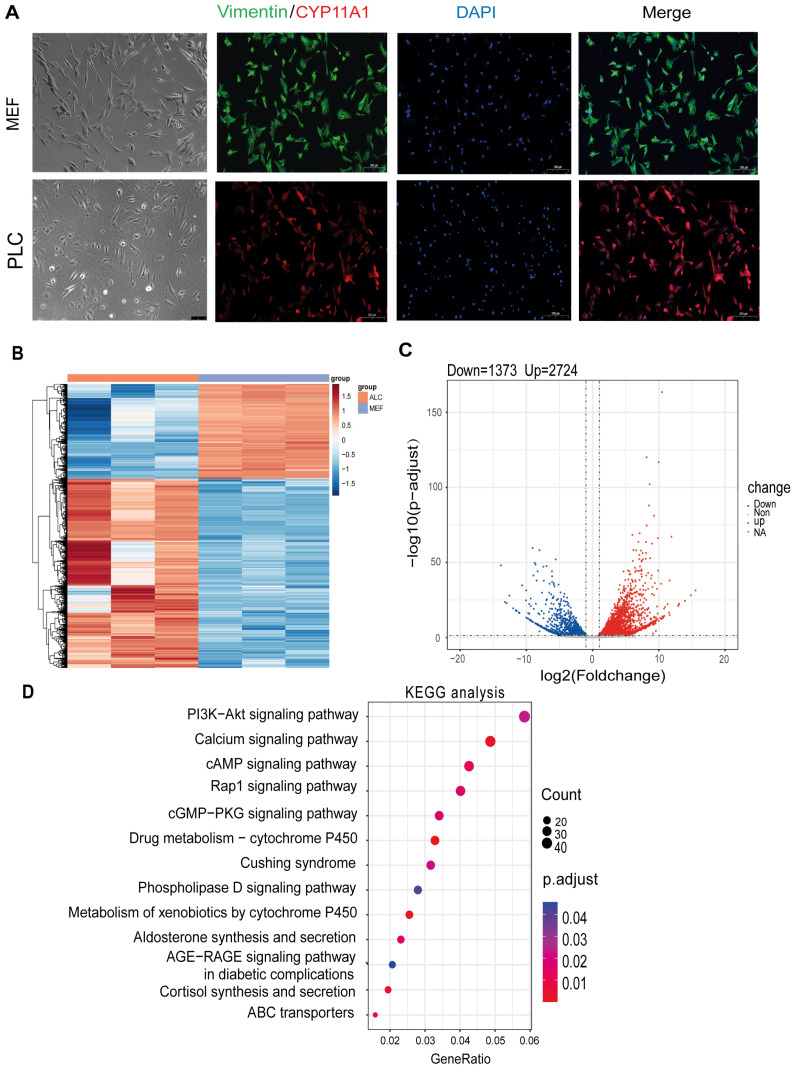
Signaling pathway differences between MEFs and PLCs as revealed by transcriptome data. (**A**) Representative light microscopy and immunofluorescence images of mouse embryonic fibroblast (MEF) and primary Leydig cell (PLC) markers. Scale bars, 200 μm; (**B**) Heatmap of differential genes in transcriptome data. Blue indicates decreased expression while red indicates increased expression; (**C**) Volcano plot of differential genes between MEFs and PLCs. Blue indicates decreased expression while red indicates increased expression; (**D**) Bubble chart of enriched Kyoto Encyclopedia of Genes and Genomes (KEGG) analysis of differential genes.

**Figure 2 pharmaceutics-15-02456-f002:**
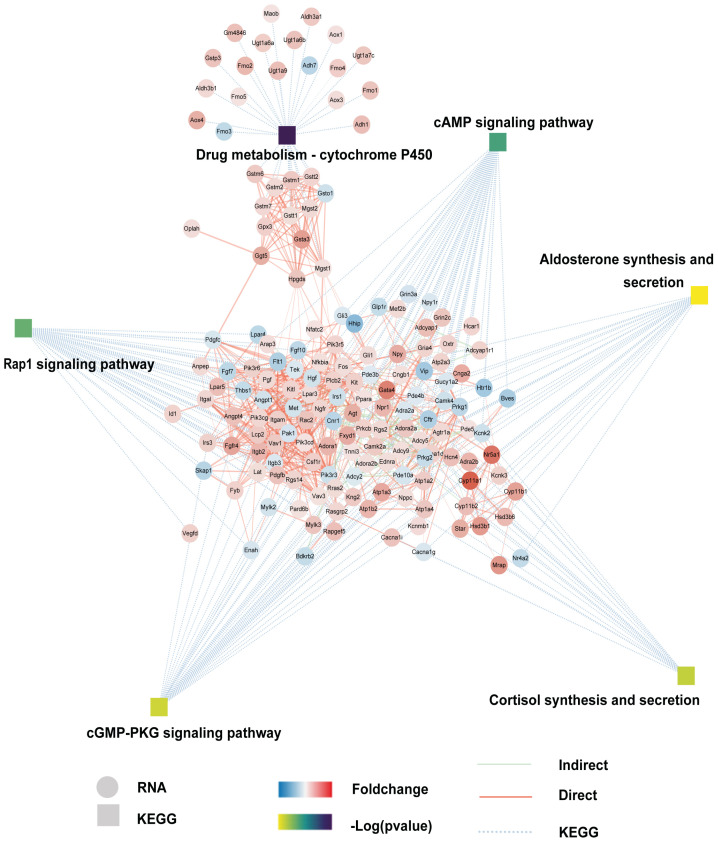
Protein–protein interaction (PPI) network map of the KEGG enriched signal pathway of differential genes.

**Figure 3 pharmaceutics-15-02456-f003:**
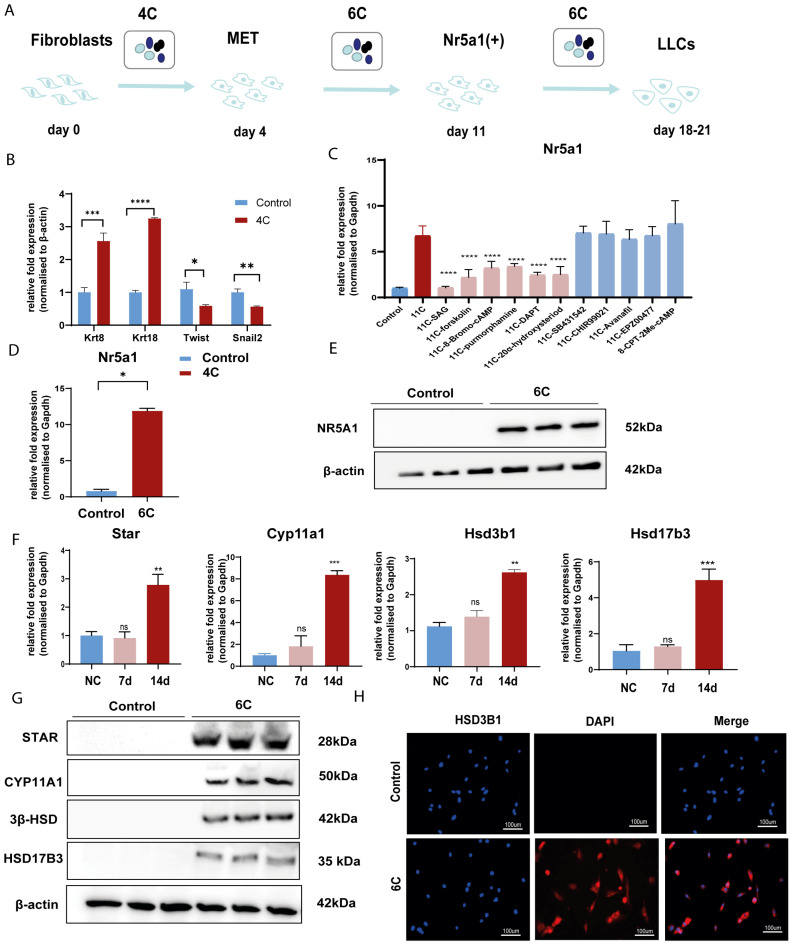
Small molecule compounds induce fibroblasts to express enzymes associated with testosterone synthesis. (**A**) Schematic diagram of fibroblast enzyme induction by small molecules; (**B**) Expression levels of HFFs fibroblast genes *Twist* and *Snail2* and epithelial gene mRNAs Krt8 and Krt 18 measured 4 days after 4C induction by qPCR; (**C**) Expression of HFFs Nr5a1 after 11C as determined by qPCR; (**D**) The expression of HFFs Nr5a1 after 7 days of treatment with 6C as determined by qPCR; (**E**) The expression of HFFs Nr5a1 after 7 days of treatment with 6C as determined by Western blot; (**F**) qPCR detection of enzyme Star, Cyp11a1, Hsd3b1, and Hsd17b3 mRNA expression in HFFs after 6C treatment for 7 days and 14 days; (**G**) Western blot of protein expression of testosterone synthesis Star, Cyp11a1, Hsd3b1, and Hsd17b3 after 14 days of 6C treatment; (**H**) Efficiency of intracellular Hsd3b1 expression in HFFs after 14 days of 6C treatment as detected by cellular immunofluorescence (n = 3, * *p* < 0.05, ** *p* < 0.01, *** *p* < 0.001, **** *p* < 0.0001 versus the control group or versus the 11C group in subfigure (**C**)).

**Figure 4 pharmaceutics-15-02456-f004:**
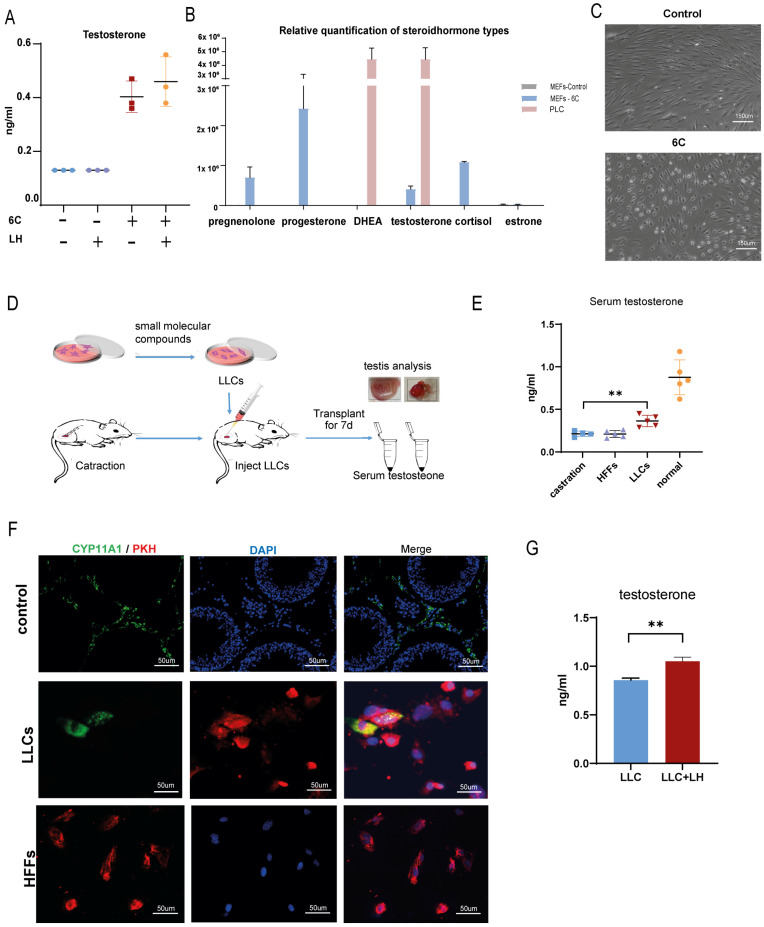
LLCs induced by small molecule compounds secreted androgens in vivo and in vitro. (**A**) LLCs and HFFs before and after LH stimulation; (**B**) LC-MS detection of steroid hormone types produced by LLCs and PLCs; (**C**) Small molecule compounds induce morphological changes in HFFs and LLCs; (**D**) Gonad deletion and cell transplantation model; (**E**) Results of the serum testosterone test for 7 days of cell transplantation in a gonad castrated mice model; (**F**) Immunofluorescence of CYP11A1 (green) expression in testis 7 days after cell transplantation in gonads; red fluorescence showed the LLCs and HFFs labeled with PKH26 transplanted into the tunica tunica of the testis, normal testicular tissue is shown in the control group; (**G**) Changes in serum testosterone levels of LLC-transplanted and castrated mice after stimulating with LH (n = 3, ** *p* < 0.01).

**Figure 5 pharmaceutics-15-02456-f005:**
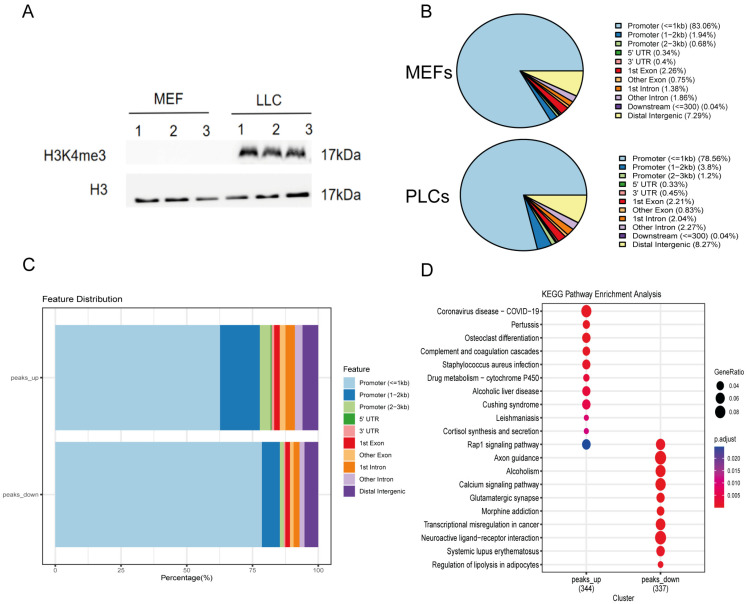
Alteration of H3K4me3 in LLCs induced by small molecule compounds. (**A**) Western blot of expression levels of MEFs and LLCs H3K4me3; (**B**) Distribution of CUT&Tag peaks of H3K4me3 across genomic regions in the second replicate of the control sample; (**C**) Annotation of the called peaks revealed H3K4me3 peaks of the second replicate; (**D**) Bubble plot showing enrichment of the differential genes in the KEGG pathways.

**Table 1 pharmaceutics-15-02456-t001:** Information on small molecular compounds.

Full Name	Functions	Source	Working Concentration	Molecular Weight	Structure
CHIR99021	GSK3 inhibition	Sigma (Kawasaki-shi, Kanagawa, Japan, Cat. No: SML1046)	10 μM	465.34	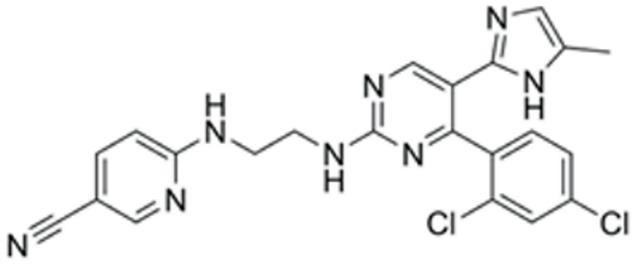
EPZ004777	DOT1L inhibition	MCE (Monmouth Junction, NJ, USA, Cat. No: HY-15227)	10 μM	539.67	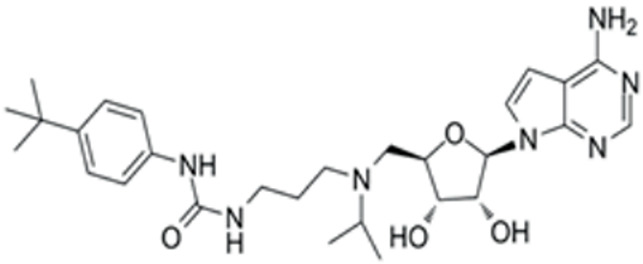
SB431542	TGFβ inhibition	Selleck (Houston, TX, USA, Cat. No: S1067)	2 μM	384.4	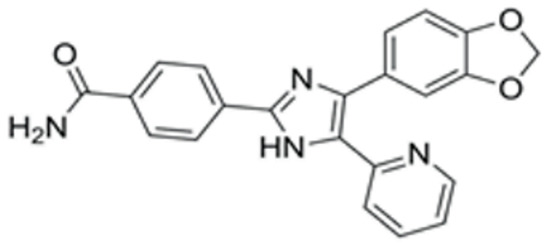
SAG	Desert Hedgehog activation	Selleck (Cat. No: S7779)	0.2 μM	490.06	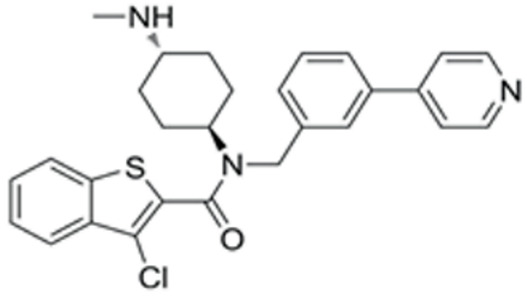
Forskolin	cAMP/PKA activation	Sigma (Cat. No: F6886)	10 μM	410.5	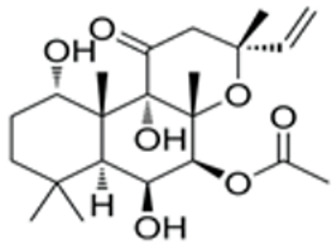
8-Bromo-cAMP	cAMP/PKA activation	Sigma (Cat. No: B7880)	1 μM	430.08	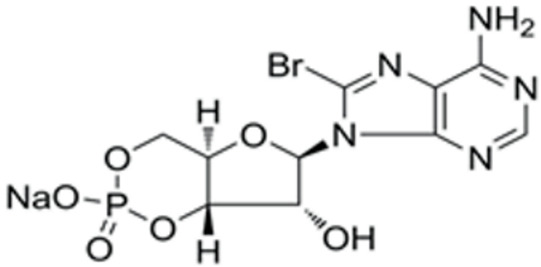
Purmorphamin	Desert Hedgehog activation	Selleck (Cat. No: S3042)	1 μM	520.6	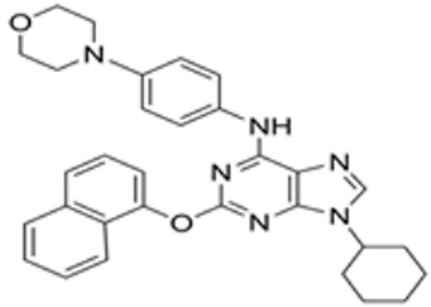
DAPT	Notch inhibition	Sigma (Cat. No: D5942)	1 μM	432.46	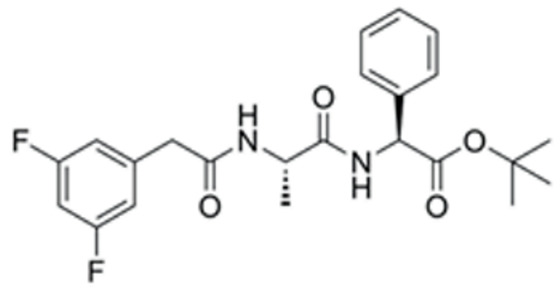
20α-hydroxy-cholesterol	Hedgehog activation	MCE (Cat. No: HY-123 16)	2 μM	402.65	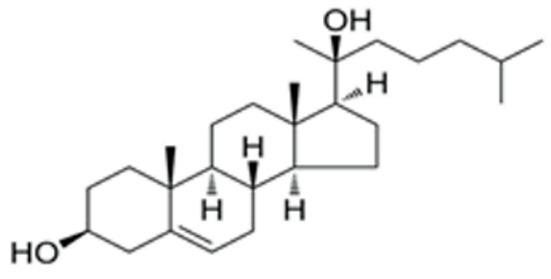
8-CPT-2Me-cAMP	Rap1 activation	MCE (Cat. No: HY-1075 43)	2 μM	507.82	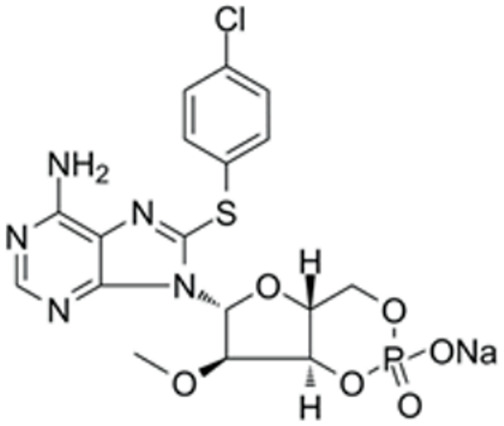
Avanafil	cGMP/PKG activation	MCE (Cat. No: HY-18252)	2 μM	483.95	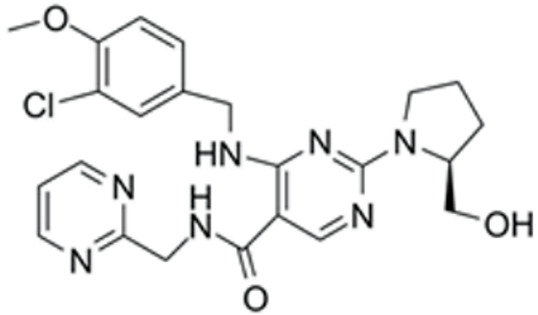

## Data Availability

The sequencing data generated in this study were uploaded to the Sequence Read Archive (SRA) database and are publicly available. The accession number is provided. Alternatively, data are available from the corresponding author upon reasonable request.

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
