# Peer review of "Small Molecule Cocktails Promote Fibroblast-to-Leydig-like Cell Conversion for Hypogonadism Therapy"

_pharmaceutics, 2023, doi:10.3390/pharmaceutics15102456_

Round 1

Reviewer 1 Report

The authors have developed an interesting study about new small molecules cocktails for hypogonadism therapy. The study is serious and rigorous but I think that it is necessary to improve the manuscript with that comments:

- There are some typographical errors: for example: for temperature should be XXoC, line 65, 67, 90…..Some chemical structures Not CO2, shoul be CO2 (subscript), in line 68, 77….

- Table 1 indicates the concentration of the different small molecules, but the text does not indicate the amount of mixture that is added to the cell cultures, nor is it indicated in the in vivo studies.

- In figure 1, I would separate the different sections to be able to see them better. For example figure 1e nothing is seen.

- Similar question in Table I: the image quality should be improved. The chemical structures don't look good.

- In the results, it should be better explained why they chose those 11 small molecules, why they begin the studies with those 4C in particular, similar to how they later explain the use of 6C.

- In some cases the nomenclature 4C and 6C can be confusing with the temperature as it has some errors where the "degrees" do not appear, as I have already indicated.

 I think that the manuscript can be accepted for its publication with these modifications.

Reviewer 2 Report

The manuscript entitled "Nanoscopic small molecules cocktails promote fibroblast-to-Leydig-like cell conversion for hypogonadism therapy" from Yuan et al involved the utilization of current small molecules as a potential cocktail of drugs for the treatment of hypogonadism.

The introduction is according to the developed topic of the manuscript, and it has updated bibliographical references to support the research.

Moreover, the methodology is related to the aim of the authors to achieve their goals.

Furthermore, the information they authors described is supported with clear and logical images and figures that summarize all the required data.

Besides, there are some points of concerns about the design of this study and the methodology they used such as:

1) Lines 221-225: The authors selected 11 small molecules for this study, but which was the characteristics or analyses they performed to select only these 11 molecules. Please clarify (Is there any assay they developed to select one over the others?)

2) Table 1 and working concentration parameters. Is not clear enough the utilization of different concentrations of each compound in the mixture, why the authors decided to use different concentrations? Did they tested if the same concentration for each compound would display the same final results?

3) Considering the structural differences among all the selected molecules, is there any conclusions regarding this important point?

Finally, I would like to invite the authors to include the abbreviation list of words at the end of this manuscript.

Reviewer 3 Report

* The authors investigated the effects of Nanoscopic small molecule cocktails on the fibroblasts' conversion into Leydig-like cells. This study is interesting and has a good impact on the hypogonadism therapy. I have some issues that need to be responded to.

* The abstract and introduction are well-written and informative.

* In the materials and methods section, Please, mention all catalog numbers for all used chemicals and kits.

* Line 101: Please, include all used primary antibodies in a table containing their catalog numbers and sources.

* Line 119: Please, include all used primers in a table.

* Line 125: Please, include all used primary antibodies in a table containing their catalog numbers and sources.

* Line 301: Please, include all the figures within the text following the journal instructions. 

* Table 1: Please, reformat according to the journal instructions.

* Regarding western blot raw images, how did you determine the molecular weight of the detected protein?

* Please, revise the manuscript with a specialized native English editing service.

Reviewer 4 Report

The present study describes nanoscopic molecules cocktails that promote fibroblast-to Leydig-like cell conversion for hypogonadism therapy. The work is appealing and well-conducted. Please consider the following observations.

  1. Please revise and correct English grammar. 
  2. Please review the title of the study. The term nanoscopic does not fit the pharmaceutical cocktail under study. 
  3. Please include tables and figures within the text of the manuscript. 
  4. Although the authors propose the in-vitro generation of Leydig cells, it is necessary to include a study to investigate the biocompatibility of the cocktail under study. 

Moderate editing of English grammar
